# Revealing the chirality origin and homochirality crystallization of Ag$_{14}$ nanocluster at the molecular level

Xiao-Qian Liang[1], Ying-Zhou Li[2], Zhi Wang[1], Shan-Shan Zhang[1], Yi-Cheng Liu[1], Zhao-Zhen Cao[1], Lei Feng[1], Zhi-Yong Gao[3], Qing-Wang Xue[4], Chen-Ho Tung[1] & Di Sun [1✉]

Although chirality is an ever-present characteristic in biology and some artificial molecules, controlling the chirality and demystifying the chirality origin of complex assemblies remain challenging. Herein, we report two homochiral Ag$_{14}$ nanoclusters with inherent chirality originated from identical rotation of six square faces on a Ag$_8$ cube driven by intra-cluster π···π stacking interaction between pntp$^-$ (Hpntp = $p$-nitrothiophenol) ligands. The spontaneous resolution of the racemic (SD/$rac$-Ag14a) to homochiral nanoclusters (SD/$L$-Ag14 and SD/$R$-Ag14) can be realized by re-crystallizing SD/$rac$-Ag14a in acetonitrile, which promotes the homochiral crystallization in solid state by forming C–H···O/N hydrogen bonds with nitro oxygen atoms in pntp$^-$ or aromatic hydrogen atoms in dpph (dpph = 1,6-bis(diphenylphosphino)hexane) on Ag$_{14}$ nanocluster. This work not only provides strategic guidance for the syntheses of chiral silver nanoclusters in an all-achiral environment, but also deciphers the origin of chirality at molecular level by identifying the special effects of intra- and inter-cluster supramolecular interactions.

[1] School of Chemistry and Chemical Engineering, State Key Laboratory of Crystal Materials, Shandong University, Ji'nan, People's Republic of China. [2] Shandong Provincial Key Laboratory of Molecular Engineering, Qilu University of Technology (Shandong Academy of Science), Ji'nan, People's Republic of China. [3] School of Chemistry and Chemical Engineering, Collaborative Innovation Center of Henan Province for Green Manufacturing of Fine Chemicals, Key Laboratory of Green Chemical Media and Reactions, Ministry of Education, Henan Normal University, Henan, Xinxiang, People's Republic of China. [4] Shandong Provincial Key Laboratory of Chemical Energy Storage and Novel Cell Technology, and School of Chemistry and Chemical Engineering, Liaocheng University, Liaocheng, People's Republic of China. ✉email: dsun@sdu.edu.cn

Chirality is ubiquitous in (supra)molecular structures, such as DNA in living organisms and natural products in plants, which plays a pivotal role in biological activity, catalysis, medicines, and a variety of other applications[1,2]. Although the enantioselective preparation and separation of chiral small molecules induced by chiral reagents have been well established, the realization of this process with achiral reagents still faces great challenges, especially for these higher-order motifs such as nanoclusters, nanoparticles, and supramolecules[3–14]. Recent research on the origin of chirality formed in achiral systems has led to great advances in chiral metal nanoclusters protected by achiral ligands; the origin of chirality of them has been typically classified into three types: (1) chiral arrangement of inner metal core atoms;[15,16] (2) asymmetric arrangement of surface structure to form a chiral shell;[17–21] (3) distortion or rotation induced structure chirality[22–24]. The majority of chirality typically belongs to the second type in nanoclusters because inter-ligand non-covalent interactions such as π⋯π stacking, C–H⋯π, hydrogen bond interactions etc. can effectively drive the asymmetric arrangement of surface structures[25]. While the last type is rather difficult to access in experiments due to the more subtle influencing factors. As we know, while the silver nanoclusters are more likely to adopt polyhedral structures with high symmetry, such as $Ag_{180}$, $Ag_{100}$, $Ag_{90}$, $Ag_{63}$, $Ag_{48}$, $Ag_{46}$, $Ag_{40}$, and $Ag_{38}$[26–31], those with chiral polyhedral cores, however, are rarely reported[32].

For achiral ligands protected chiral nanoclusters, a long-standing challenge lies in the fact that most of them will crystallize as racemates, and highly efficient enantioseparation technologies are thus urgently needed. To date, some enantioseparation technologies such as high-performance liquid chromatography, capillary electrophoresis and chiral ion pairs have been developed to separate enantiomers, but there are still some limitations to enantioseparation for most racemates[33–38]. For example, for some chiral molecules that cannot maintain a stable chiral conformation in solution, above enantioseparation technologies operated in solution phase will not be efficient enough. Thus, the solvent-induced crystallization resolution may serve as an upgrade alternative to achieve enantiomer separation.

In this work, we firstly isolate a racemic $Ag_{14}$ nanocluster ($[Ag_{14}(pntp)_{10}(dpph)_4Cl_2]$, **SD/rac-Ag14a**, Hpntp = p-nitrothiophenol and dpph = 1,6-bis(diphenylphosphino)hexane) in acetone/$CH_2Cl_2$, which is further subjected to a separation into their component enantiomers, **SD/L-Ag14** and **SD/R-Ag14**, by re-crystallizing **SD/rac-Ag14a** in acetonitrile. As revealed by single-crystal X-ray diffraction (SCXRD), the inherent chirality of $Ag_{14}$ nanocluster originates from identical rotational directionality of six square faces on $Ag_8$ cube, which is mainly driven by intra-cluster π⋯π stacking interaction between pntp⁻ ligands, while homochiral crystallization is promoted by C–H⋯O/N hydrogen bonds formed between lattice acetonitrile and the nitro oxygen atoms in pntp⁻ or aromatic hydrogen atoms in dpph on $Ag_{14}$ nanocluster. The enantiomeric conglomerates show mirror-imaged circular dichroism (CD) and circularly polarized luminescence (CPL) responses. These results give us precise answers at molecular-level to (i) How to desymmetrize the highly symmetrical polyhedral silver nanocluster to become a chiral one; (ii) what is responsible for the chirality of the overall molecule; (iii) what is the driving force to enantioseparate racemates during the crystallization.

## Results

### Syntheses and characterizations of SD/rac-Ag14a and SD/L/R-Ag14.
The chiral $Ag_{14}$ nanocluster was one-pot synthesized as racemates (**SD/rac-Ag14a**) using phosphine (dpph or $PPh_3$), Hpntp, and $CF_3COOAg$ with a ratio of 1:2:2 in acetone/$CH_2Cl_2$ at

room temperature. The filtrate was evaporated slowly to produce orange rod-like crystals of **SD/rac-Ag14a**. $NaBH_4$ was chosen as the reducing agent to obtain the subvalent kernel $Ag_6^{4+}$. The amount of triethylamine has a nontrivial influence on the dimensions of single crystals. The 0–100 μL triethylamine has been attempted in this reaction system with 10 μL as an interval and larger single crystals can be formed in the case of 40 μL triethylamine. Furthermore, the other two bases such as NaOH and N,N,N′,N′-Tetramethylethylenediamine (TMEDA) were also tried in the synthesis of $Ag_{14}$ nanocluster. The TMEDA can also work as triethylamine, whereas the NaOH is not the case; this is probably due to the slower reduction kinetics of $NaBH_4$ in NaOH, which impedes the formation of $Ag_{14}$ nanocluster. The growth of large-sized single crystals of the enantiopure **SD/L/R-Ag14** is difficult but necessary for collecting their more reliable solid-state CD and CPL signals. The detailed synthetic route to **SD/Ag14** is shown in the Supplementary Fig. 1.

Separation of the racemates into single enantiomer by spontaneous resolution is very hard to achieve for metal nanoclusters because of inherent difficulty during crystallization. Therefore we try to enantioseparate **SD/rac-Ag14a** by tuning the solvent system. Initially, **SD/rac-Ag14a** crystallized in the $C2/c$ space group in acetone/$CH_2Cl_2$. In comparison, the enantiomeric **SD/L-Ag14** and **SD/R-Ag14** can be directly obtained in MeCN/$CH_2Cl_2$ with a crystallographic space group of $P2_12_12_1$. They possess completely different morphology (block) with respect to **SD/rac-Ag14a** (rod) (Supplementary Fig. 2). We also found that re-crystallization of **SD/rac-Ag14a** in MeCN can also produce enantiomeric **SD/L-Ag14** and **SD/R-Ag14**, which can convert back to **SD/rac-Ag14a** through re-crystallization in acetone (Fig. 1). All these results suggest the important role of solvent in regulating the crystallization of chiral silver nanoclusters.

Apart from SCXRD, **SD/rac-Ag14a, SD/L-Ag14, and SD/R-Ag14** were thoroughly characterized by energy dispersive spectroscopy (EDS), nuclear magnetic resonance spectroscopy (NMR), electrospray ionization mass spectrometry (ESI-MS), ultraviolet-visible (UV-Vis), powder X-ray diffraction (PXRD) and infrared spectroscopy (IR). These characterizations, additional structural graphics, as well as crystallography-related information are collected in the Supplementary Figs. 3–23, Supplementary Tables 1–6. Moreover, density functional theory (DFT) calculations were also performed to understand the electronic structure and optical properties.

### Crystal structure of SD/rac-Ag14a.
SCXRD analysis determined the molecular formula of $Ag_{14}$ nanocluster to be $[Ag_{14}(pntp)_{10}(dpph)_4Cl_2]$. **SD/rac-Ag14a** crystallizes in the monoclinic space group $C2/c$ with a complete cluster in the asymmetric unit (Supplementary Table 1). The symmetry breaking of $Ag_{14}$ nanocluster with alternating packing of opposite handedness ones finally results in racemization, therefore the unit cell of **SD/rac-Ag14a** contains two equivalent enantiomeric nanoclusters, **SD/L-Ag14** and **SD/R-Ag14**. The $Ag_{14}$ nanocluster is composed of 14 silver atoms, 10 pntp⁻ ligands, 4 dpph ligands, and two chlorides (Fig. 2a). The EDS mapping clearly demonstrate the uniform distribution of elements in the nanoclusters and the presence of Cl (Supplementary Fig. 3). The $Ag_{14}$ nanocluster can be regarded as a core-shell structure, with the octahedral $Ag_6$ as the core surrounded by a twisted $Ag_8$ cube as the shell. **SD/rac-Ag14a** is a neutral cluster as indicated by ESI-MS discussed below, thus two additional electrons should be injected into it, making the central $Ag_6$ octahedron +4 charged[39]. The distorted $Ag_8$ cube breaks the overall $O_h$ symmetry, and therefore endows the nanocluster with chirality (Fig. 2a). In addition, the weak π⋯π interactions between pntp⁻ ligands with

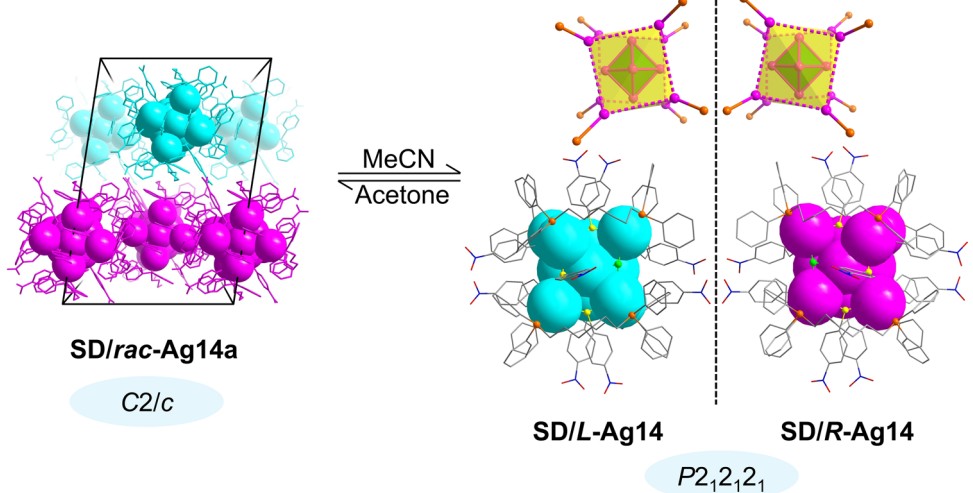

**Fig. 1 Schematic illustration of the spontaneous resolution process.** Solvent-induced spontaneous resolution of **SD/*rac*-Ag14a** to **SD/*L*-Ag14** and **SD/*R*-Ag14**. Color labels: cyan and pink, Ag; yellow, S; orange, P; green, Cl; gray, C; blue, N; red, O. All H atoms are omitted for clarity.

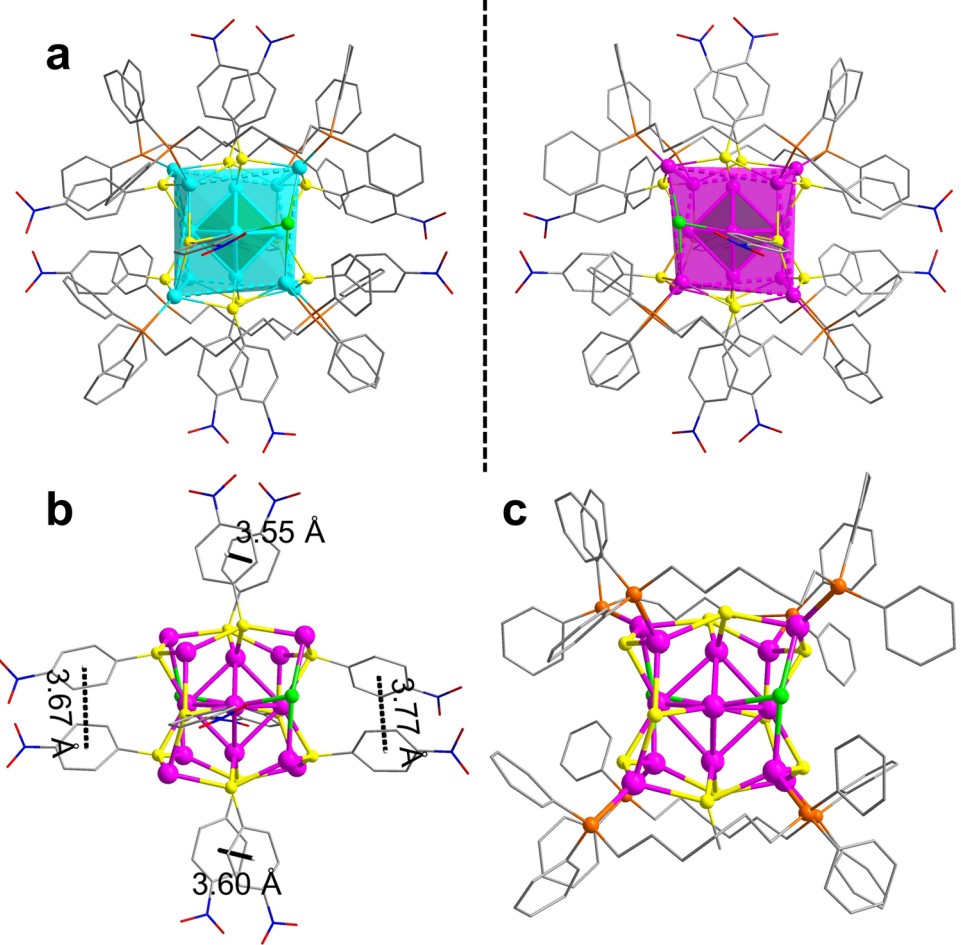

**Fig. 2 Molecular structures of SD/*rac*-Ag14a. a** Mirror symmetry of the enantiomers with $Ag_6$ core and $Ag_8$ shell depicted as colored polyhedra. **b** The coordination mode of pntp⁻ (Hpntp = *p*-nitrothiophenol) and the π···π interaction (black dashed lines) between pntp⁻ ligands in **SD/*rac*-Ag14a. c** The coordination mode of dpph (dpph = 1,6-bis(diphenylphosphino)hexane). Color labels: cyan and pink, Ag; yellow, S; orange, P; green, Cl; gray, C; blue, N; red, O. All H atoms are omitted for clarity.

the centroid-centroid distance ranging from 3.55 to 3.77 Å are clearly observed (Fig. 2b)[40,41].

**SD/*L*-Ag14** was chosen as the representative for more detailed structural analysis. The Ag···Ag distance in $Ag_6$ fall in the range of

2.79–2.89 Å (Supplementary Table 2), indicating the presence of argentophilic interactions[42–44]. The coordination modes of pntp⁻ and dpph are shown in Figs. 2b, c. The ten pntp⁻ ligands ride on ten edges of $Ag_8$ cube, whereas the remaining two edges are

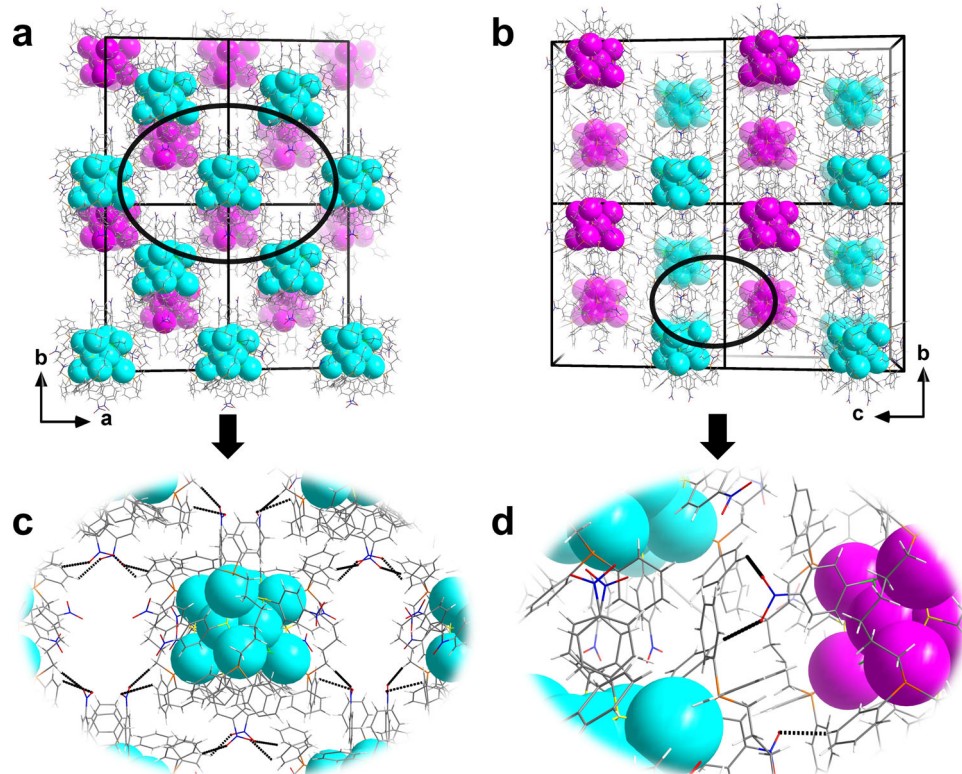

**Fig. 3 Packing structure of SD/*rac*-Ag14a. a** Packing structure of enantiomers in a 2 × 2 × 2 cell viewed along the *c* axis. **b** Packing structure of enantiomers in a 2 × 2 × 2 cell viewed along the *a* axis. **c** Zoom-in inter-cluster weak interactions in the same layer. **d** Zoom-in inter-cluster weak interactions between neighboring layers. Color labels: cyan and pink, Ag; yellow, S; orange, P; gray, C; green, Cl; blue, N; red, O. white, H. Black dashed lines indicate the weak interactions.

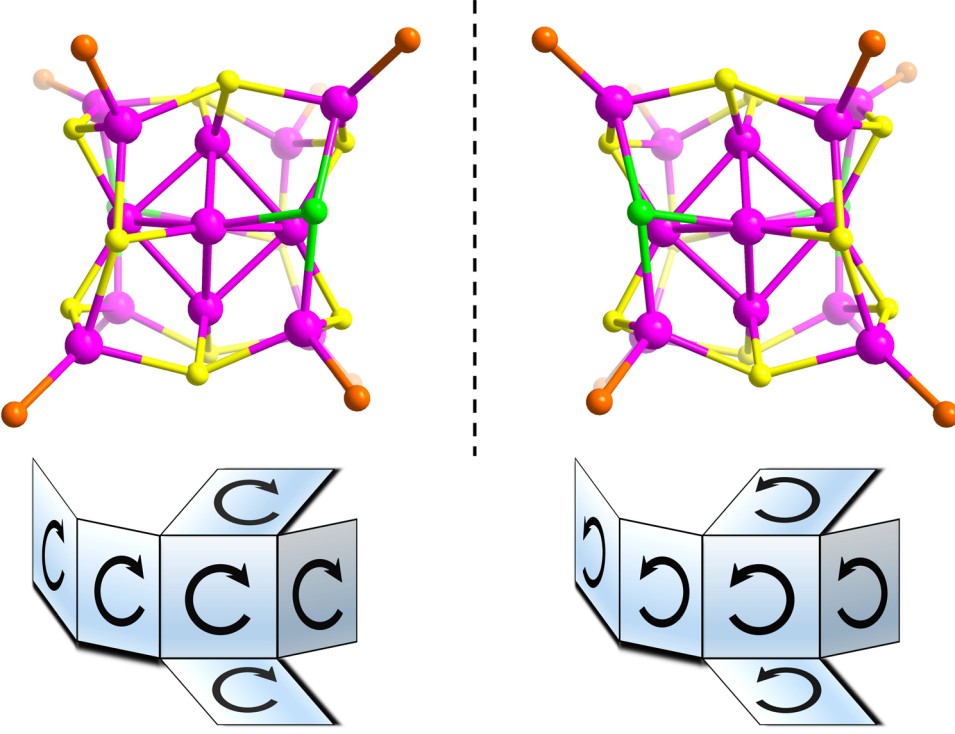

**Fig. 4 Schematic representation of rotation of all square faces on Ag$_8$ cube.** The rotational direction of six square faces of Ag$_8$ cube in **SD/*L*-Ag14** (left) and **SD/*R*-Ag14** (right) nanoclusters. Color labels: pink, Ag; yellow, S; orange, P; green, Cl. All C, N, O, and H atoms are omitted for clarity.

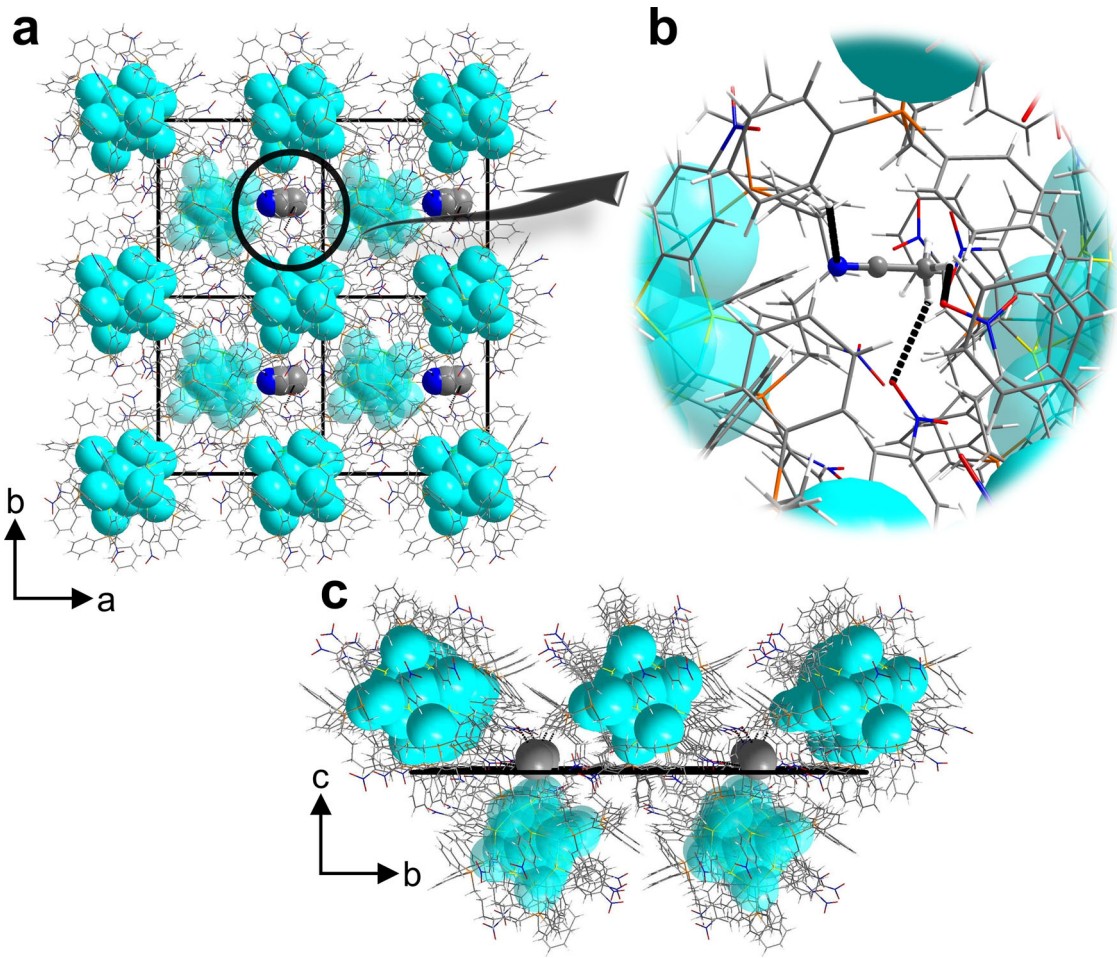

**Fig. 5 Packing structure of SD/*L*-Ag14. a** Packing structure of **SD/*L*-Ag14** in a 2 × 2 × 2 cell viewed along the *c* axis. **b** Zoom-in inter-cluster $C_{acetonitrile}$–H···$O_{nitro}$ and $C_{dpph}$–H···$N_{acetonitrile}$ hydrogen bonds in the same layer (dpph = 1,6-bis(diphenylphosphino)hexane). **c** Packing structure of **SD/*L*-Ag14** in a 2 × 2 × 2 cell viewed along the *a* axis. Color labels: cyan, Ag; yellow, S; orange, P; gray, C; green, Cl; blue, N; red, O; white, H. Black dashed lines indicate the weak interactions.

ridden by two Cl⁻ ions. Each pntp⁻ ligand adopts a $\mu_3$ bridging mode to bind two silver atoms on an edge and one apical silver atom from $Ag_6$ octahedron (Fig. 2b). The $Ag_{octahedron}$–S bond lengths lie in the range of 2.48–2.55 Å and the $Ag_{cube}$–S bond lengths in the range of 2.53–2.70 Å (Supplementary Table 2). The S–Ag–S bond angles lie in the range of 86.20–120.04° (Supplementary Table 2). Four of twelve edges of $Ag_8$ cube are additionally bridged by four $\mu_2$-dpph ligands (Ag–P: 2.40–2.47 Å, Supplementary Table 2). The steric hindrance of dpph impedes the full coordination of large pntp⁻ ligand at all edges of $Ag_8$ cube and two of them are bound by small Cl⁻ ion instead. The coordination behavior of Cl⁻ ion is quite similar to that of S atom of pntp⁻ ligand (Ag-Cl: 2.61–2.83 Å). Of note, the ³¹P NMR spectrum shows eight different ³¹P chemical shifts with nearly identical integration values ranging from −1.79 to 2.02 ppm in the $CD_2Cl_2$, which suggests all four dpph ligands in **SD/*L*-Ag14** and **SD/*R*-Ag14** locate in completely different chemical environments, ruling out any symmetry for the overall structure in solution (Supplementary Fig. 4).

Based on the above crystallographic analyses, we make two hypotheses responsible for the origin of chirality: (i) the dpph immobilizes the asymmetric arrangement of vertex Ag atoms of $Ag_8$ cubic shell; (ii) the locking of identical rotational directionality (Supplementary Fig. 6) of six square faces on $Ag_8$ cube by the π···π interaction between the pntp⁻ ligands. To verify the above hypotheses, the dpph ligand was replaced by $PPh_3$ under similar

synthetic conditions and another similar racemic nanocluster **SD/*rac*-Ag14b** was isolated. Some structural diagrams and crystallographic tables for it are shown in Supplementary Information. A detailed analysis of **SD/*rac*-Ag14b** revealed that the π···π interaction between the pntp⁻ ligands and the identical rotational directionality of six square faces on $Ag_8$ cube still exist (Supplementary Fig. 7), even in the absence of the immobilization of bridging dpph ligands, which means the factor (ii) may dominate the symmetry breaking of $Ag_8$ cube.

In the 2 × 2 × 2 cell of **SD/*rac*-Ag14a**, the $Ag_{14}$ nanoclusters with opposite chirality alternatively align in the planes parallel to crystallographic *ab* plane (Fig. 3a, b), causing the final racemic crystals. The inter-cluster $C_{dpph}$–H···$O_{nitro}$ hydrogen bonds (2.53–2.59 Å; Fig. 3c) consolidate the packing of homochiral $Ag_{14}$ nanoclusters in the same layer. The homochiral layers composed of **SD/*L*-Ag14** or **SD/*R*-Ag14** alternate in an AB-type packing fashion along the *c* axis through similar hydrogen bonds (2.35–2.84 Å; Fig. 3d).

**Crystal structures of SD/*L*-Ag14 and SD/*R*-Ag14.** To realize the homochiral crystallization of **SD/*rac*-Ag14a** for the following chiroptical studies, we changed the acetone/$CH_2Cl_2$ to MeCN/$CH_2Cl_2$ to get enantiomeric **SD/*L*-Ag14** and **SD/*R*-Ag14**. The bulk samples of them were collected by manually picking single crystals then checking their configurations one by one by SCXRD.

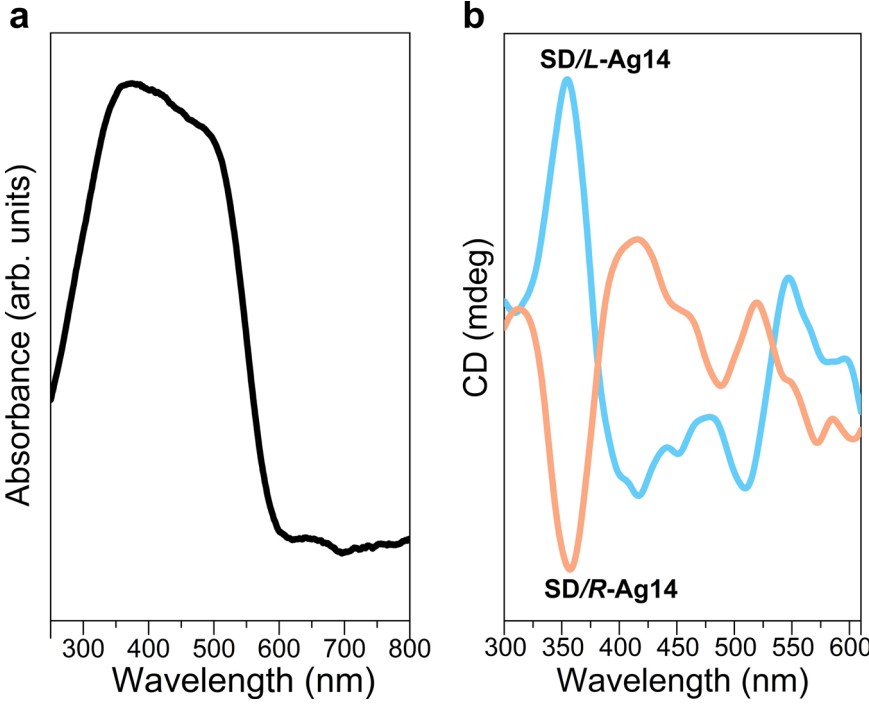

**Fig. 6 The electronic spectra of of SD/_L_-Ag14 and SD/_R_-Ag14 in the solid state. a** UV-Vis absorption spectrum of racemic conglomerates **SD/_L_-Ag14** and **SD/_R_-Ag14** in solid state. **b** Circular dichroism spectra of the **SD/_L_-Ag14** and **SD/_R_-Ag14** in solid state.

Crystallographic analysis showed that the unit cell indeed contains homochiral nanoclusters, and additionally co-crystallized MeCN was identified, thus the molecular formula of enantiomeric Ag$_{14}$ nanoclusters is $[Ag_{14}(pntp)_{10}(dpph)_4Cl_2]\cdot MeCN$ (**SD/_L_-Ag14** or **SD/_R_-Ag14**). **SD/_L_-Ag14** and **SD/_R_-Ag14** both crystallized in an orthorhombic space group of $P2_12_12_1$ (Supplementary Table 1). All 40 crystals grown in one beaker from one-batch synthesis were analyzed by SCXRD and the ratio of **SD/_L_-Ag14** and **SD/_R_-Ag14** is 21: 19, indicating the spontaneous enantiomer resolution occurs (Supplementary Table 3).

To obtain deep insights into the chiral origin of **SD/_L_-Ag14** and **SD/_R_-Ag14**, the rotation direction of six square faces of Ag$_8$ cube are scrutinized by unfolding them in a two-dimensional plane (Fig. 4). It can be clearly seen that every square face of the Ag$_8$ cube shows the same rotation direction in **SD/_L_-Ag14** or **SD/_R_-Ag14**. Such rotational configuration is locked by the $\pi\cdots\pi$ interaction between the pntp$^-$ ligands to maintain such chirality in solid state. In both, codirectional face-rotation of polyhedron generates the inherent chirality, which is the origin of chirality of Ag$_{14}$ nanocluster.

Why does the CH$_3$CN has such magic to homochiral crystallization of Ag$_{14}$ nanocluster? We chose the packing structure of **SD/_L_-Ag14** as a representative for the following analysis. In a $2 \times 2 \times 2$ cell of **SD/_L_-Ag14**, the homochiral Ag$_{14}$ nanoclusters are self-assembled into one-dimensional columnar arrangements along _b_ axis (Fig. 5a, c). Such assembly is mainly driven by hydrogen bonds which contain non-classic but important C$_{acetonitrile}$–H$\cdots$O$_{nitro}$ and C$_{dpph}$–H$\cdots$N$_{acetonitrile}$ hydrogen bonds (C$_{acetonitrile}$–H$\cdots$O$_{nitro}$ and C$_{dpph}$–H$\cdots$N$_{acetonitrile}$: 2.44–2.67 Å; Fig. 5b).

**Mass spectrometry of SD/_L_-Ag14 and SD/_R_-Ag14.** As we know, ESI-MS is a powerful analyzing tool to verify the chemical composition, charge state and the stability of metal nanoclusters[45–48]. The positive-ion ESI-MS of **SD/_L_-Ag14** and **SD/_R_-Ag14** was measured by dissolving them in methanol/

CH$_2$Cl$_2$. As shown in Supplementary Fig. 8, there are three most prominent peaks in the mass-to-charge ratio (_m/z_) ranges of 4415–4435 (A), 4955–4972 (B), and 5368–5388 (C), respectively. Each group of isotopic peaks of them contains monovalent species, where the most dominant peak (B) is intact molecular ion species but with one sodium attachment. The peak B is located at 4964.3536 and can be assigned to $[NaAg_{14}(pntp)_{10}(dpph)_4Cl_2]^+$ (**SD/_L_-Ag14** + Na$^+$; Calcd. 4964.3537), therefore, the Ag$_{14}$ nanocluster is a 2-electron superatom system. The three species were identified based on the perfect matching of observed and simulated isotopic distributions. All these assigned formulas are listed in Supplementary Table 4.

**The solid-state UV-Vis and CD spectra of SD/_L_-Ag14 and SD/ _R_-Ag14.** The UV-Vis diffuse reflection spectra of **SD/_rac_-Ag14a**, **SD/_L_-Ag14**, and **SD/_R_-Ag14** show the same absorption peaks in solid state at room temperature, indicating that the cluster packing of nanoclusters does not essentially contribute to the optical absorbance. The broad absorption at 300–600 nm (Fig. 6a), consistent with its orange color of crystals, probably involve the ligand-to-ligand charge transfer (LLCT) mixed with small proportion of metal-to-ligand charge transfer (MLCT) transitions (_vide infra_). The CD and corresponding absorption spectra of **SD/_L_-Ag14** and **SD/_R_-Ag14** were also collected at 300–600 nm in solid state (Fig. 6b and Supplementary Fig. 9). The CD spectra of **SD/_L_-Ag14** and **SD/_R_-Ag14** are nearly perfect mirror images of each other, confirming their enantiomeric relationship; the broad absorption peaks are split into several peaks by the Cotton effect, including a major peak at around 355 nm and other less resolved peaks ranging from 400 to 500 nm.

To further understand the origin of chirality, taking **SD/_L_-Ag14** as an example, its metal core after DFT optimization in gas phase remains almost intact and thus its calculated CD spectrum can be used to help interpret the origin of its chirality. The calculated CD spectrum of **SD/_L_-Ag14** matches fairly well with its experimental spectrum in the region below 500 nm, given the

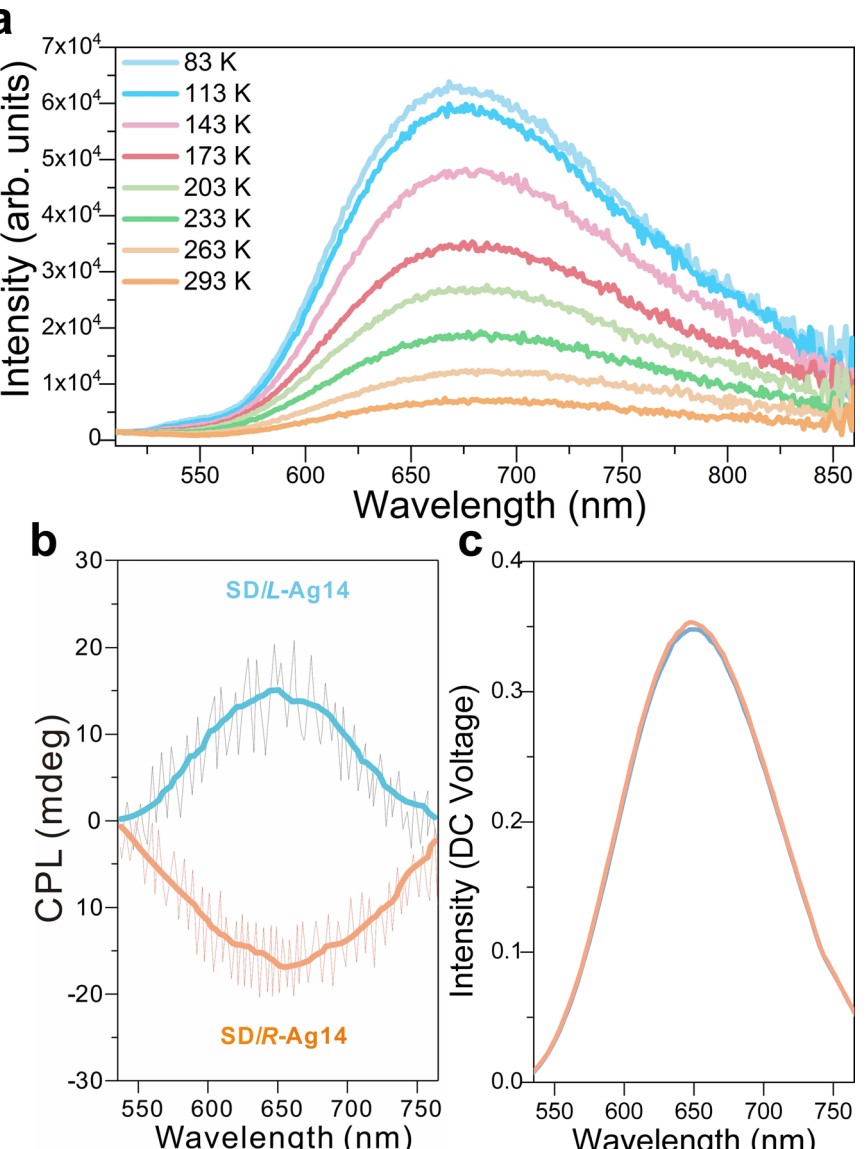

**Fig. 7 Luminescence spectra of SD/L-Ag14 and SD/R-Ag14. a** Variable-temperature emission spectra of racemic conglomerates **SD/L-Ag14** and **SD/R-Ag14** in the solid state under the excitation of 365 nm. **b** Circularly polarized luminescence spectra of **SD/L-Ag14** or **SD/R-Ag14** in the solid state under the excitation of 365 nm. **c** Total fluorescence intensity spectra of **SD/L-Ag14** (blue solid line) or **SD/R-Ag14** (orange solid line) in the solid state under the excitation of 365 nm.

blue-shift in its calculated UV-Vis spectrum (Supplementary Fig. 10). The broad calculated absorption peak was also split into three peaks by Cotton effect, including two positive peaks at 355 nm and 457, respectively, and one broad negative peak at 400 nm. All the CD responses are mainly associated with the LLCT transitions, despite the small proportion of metal-to-metal charge transfer (MMCT) and MLCT transitions, as illustrated in its UV-Vis absorption analysis (Supplementary Fig. 14 and Supplementary Table 6). This suggests that the chiral response in the Ag$_{14}$ nanocluster mainly originates from the asymmetrical arrangement of the surface ligands, which is induced by non-covalent interactions (mainly the π···π stacking interaction) between pntp$^-$ ligands.

**Photoluminescent and CPL properties.** The photoluminescent properties of **SD/L-Ag14** and **SD/R-Ag14** were examined in solid state and exhibited maximum emission around at 680 nm under the excitation of 365 nm at room temperature (Fig. 7a). The

variable-temperature fluorescence spectrum of **SD/rac-Ag14a** (Supplementary Fig. 11) is almost identical to those of **SD/L-Ag14** and **SD/R-Ag14**. The luminescence lifetimes are also determined in microsecond regime at 83 and 293 K (Supplementary Fig. 12), indicating that excited states exist in a triplet state. Upon cooling from 293 to 83 K, the emission maximum slightly blue-shifted from 680 to 668 nm ($\Delta\lambda = 12$ nm) with a 9-fold enhancement in intensity, which is mainly ascribed to the reduced nonradiative decay by limiting the intramolecular rotations and vibrations at low-temperature. Furthermore, the long-wavelength emissions with microsecond lifetime are predominantly originated from $^3$MLCT transitions[5,6]. The CPL properties of **SD/L-Ag14** and **SD/R-Ag14** were also explored. As shown in Fig. 7b, c, enantiomers display highly symmetric CPL patterns in the 550–750 nm range under the excitation of 365 nm at room temperature. The dissymmetry g-factor ($g_{lum}$) was calculated to be and its value $2.97 \times 10^{-3}$ at 650 nm by $g_{lum} = 2 \times$ [CPL/(32 980/ln10)]/DC, which is comparable to those of the reported metal nanoclusters[49,50].

## a

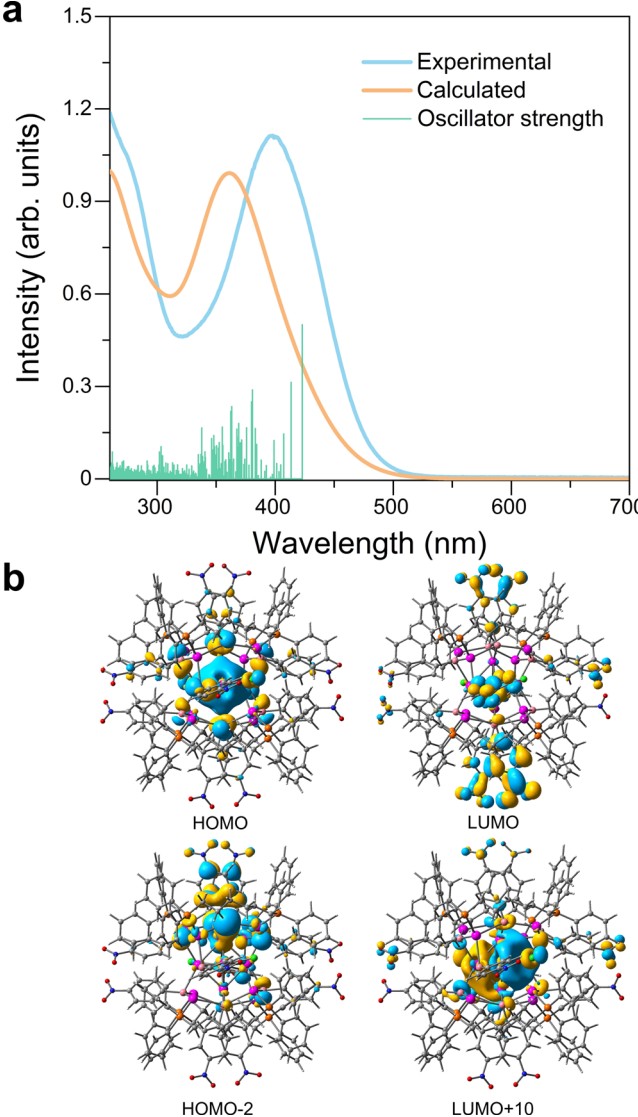

## b

HOMO

LUMO

HOMO-2

LUMO+10

**Fig. 8 Time-dependent density functional theory calculations on SD/L-Ag14. a** Experimental and time-dependent density functional theory calculated UV-Vis absorption spectra of **SD/Ag14. b** The calculated HOMO, LUMO, HOMO-2, and LUMO + 10 orbitals of **SD/L-Ag14**.

**Density functional theory calculation**. The UV-Vis spectrum of **SD/Ag14** in $CH_2Cl_2$ shows a strong and broad peak at 398 nm (Supplementary Fig. 13). Time-dependent DFT calculations on **SD/L-Ag14** were performed to better understand its electronic structure (Supplementary Data 1). As shown in Fig. 8, the calculated electronic spectrum also exhibits one broad peak centered at 358 nm, which most probably corresponds to the experimental peak at 398 nm although it is blue-shifted by 40 nm. The HOMO orbital of **SD/L-Ag14** has an obvious superatom S character, consistent with its identity as a 2-electron superatom. The other occupied orbitals from HOMO-1 to HOMO-8, and even those with lower energy such as HOMO-12 and HOMO-14, mainly comprises of the p orbitals of S and C atoms in pntp⁻ ligands. The unoccupied orbitals from LUMO to LUMO + 9 are solely contributed by the $\pi^*$ orbitals of C = C and N = O bonds in the pntp⁻ ligands. The LUMO + 10, LUMO + 11, LUMO + 12 orbitals are essentially P superatom orbitals (Supplementary Fig. 14). The calculated absorption at 358 nm was mainly contributed by the transitions from the occupied orbitals (HOMO to

HOMO-8) to unoccupied orbitals (LUMO to LUMO + 12). Consequently, the electronic transitions contributed to the calculated absorption peak are mainly of LLCT character, such as HOMO-2→LUMO + 2/LUMO + 3/LUMO + 7 and HOMO-3/HOMO-4/HOMO-6→LUMO; apart from these, some MLCT (metal-to-ligand charge transfer) and MMCT (metal-to-metal charge transfer) transitions also contributed, as represented by HOMO → LUMO/LUMO + 1/LUMO + 2 and HOMO → LUMO + 10/LUMO + 11/LUMO + 12, respectively.

## Discussion

In conclusion, we isolated a racemic Ag14 nanocluster (**SD/rac-Ag14a**) protected by achiral ligands of pntp⁻ and dpph in acetone/$CH_2Cl_2$, and further separated it into their component enantiomers by re-crystallizing **SD/rac-Ag14a** in acetonitrile. The inherent chirality of Ag14 nanocluster is induced by identical rotational directionality of six square faces of Ag8 cube, which is locked by π···π interaction between pntp⁻ ligands, maintaining the chirality in solid state. Homochiral crystallization is promoted by C–H···O/N hydrogen bonds formed between lattice acetonitrile and the nitro oxygen atoms in pntp⁻ or aromatic hydrogen atoms in dpph on Ag14 nanocluster. The enantiomeric conglomerates show mirror-imaged CD and CPL responses. This work not only presents an approach for the synthesis and enantioseparation of face-rotating induced chiral silver nanoclusters protected by achiral ligands, but also provides a deeper insight into the origin and spontaneous resolution of chirality at molecular level.

## Methods

**Synthesis of SD/rac-Ag14a.** CF₃COOAg (11.0 mg, 0.05 mmol), and 1,6-bis(diphenylphosphino)hexane (11.4 mg, 0.025 mmol) were mixed in 6 mL of acetone/$CH_2Cl_2$ (v:v = 4:2) followed by addition of Hpntp (7.8 mg, 0.05 mmol). After stirring (1000 rpm) for 5 min, 40 μL trimethylamine was added to the above solution. After stirring for 20 min, 0.5 mL of EtOH solution of NaBH₄ (4 mg/mL) was added and the resulting mixture was further stirred (1000 rpm) at room temperature for 6 h, during which the color rapidly changed from yellow to black. The yellow filtrate was left to stand in the dark at room temperature. After slow evaporation for 14 days, yellow block crystals of **SD/rac-Ag14a** were collected and washed with ethanol (EtOH). Yield: 5.0 mg (34 %). Selected IR peaks (cm⁻¹) of **SD/rac-Ag14a**: 1568 (s), 1500 (s), 1473 (m), 1432 (m), 1328 (s) 1178 (m), 1083 (s), 848 (s), 734 (s), 689 (s), 508 (s). Yield: 5.0 mg (34 %).

**Synthesis of racemic conglomerates SD/L-Ag14 and SD/R-Ag14.** CF₃COOAg (11.0 mg, 0.05 mmol), and 1,6-bis(diphenylphosphino)hexane (11.4 mg, 0.025 mmol) were mixed in 6 mL of acetonitrile/$CH_2Cl_2$ (v:v = 4:2) followed by addition of Hpntp (7.8 mg, 0.05 mmol). After stirring (1000 rpm) for 5 min, 40 μL trimethylamine was added to the above solution. After stirring for 20 min, 0.5 mL of EtOH solution of NaBH₄ (4 mg/mL) was added and the resulting mixture was further stirred (1000 rpm) at room temperature for 6 h, during which the color rapidly changed from yellow to black. The orange filtrate was left to stand in the dark at room temperature. After slow evaporation for 14 days, orange block crystals of racemic conglomerates **SD/L-Ag14** and **SD/R-Ag14** were collected and washed with ethanol (EtOH). Yield: 4.5 mg (30 %). In addition, we also found the re-crystallization of **SD/rac-Ag14a** in acetonitrile can also obtain racemic conglomerates **SD/L-Ag14** and **SD/R-Ag14**, which can return back to **SD/rac-Ag14a** through re-crystallizing in acetone.

**Synthesis of SD/rac-Ag14b.** CF₃COOAg (11.0 mg, 0.05 mmol), and PPh₃ (6.5 mg, 0.025 mmol) were mixed in 6 mL of MeOH/$CH_2Cl_2$ (v:v = 4:2) followed by addition of Hpntp (7.8 mg, 0.05 mmol). After stirring (1000 rpm) for 5 min, 40 μL trimethylamine was added to the above solution. After stirring for 20 min, 0.5 mL of EtOH solution of NaBH₄ (4 mg/mL) was added and the resulting mixture was further stirred (1000 rpm) at room temperature for 6 h, during which the color rapidly changed from yellow to black. The yellow filtrate was left to stand in the dark at room temperature. After slow evaporation for 14 days, yellow block crystals of **SD/rac-Ag14b** were collected and washed with ethanol (EtOH). Yield: 2.0 mg (13 %).

## Data availability

The data that support the findings of this study are available within the article and its Supplementary Information files. Other relevant data are available from the

corresponding author upon reasonable request. The raw data for the computational calculations are provided with Supplementary Data 1. The X-ray crystallographic coordinates for structures reported in this article have been deposited at the Cambridge Crystallographic Data Centre, under deposition numbers CCDC: 2070983-2070986 for **SD/*rac*-Ag14a, SD/*L*-Ag14, SD/*R*-Ag14, and SD/*rac*-Ag14b**. These data can be obtained free of charge from the Cambridge Crystallographic Data Centre via www.ccdc.cam.ac.uk/data_request/cif.

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

## Acknowledgements

This work was financially supported by the National Natural Science Foundation of China (Grant Nos. 91961105, 21822107, 22001139, and 21827801), the Natural Science Foundation of Shandong Province (Nos. ZR2020ZD35, ZR2019ZD45, ZR2019BB058, JQ201803, and ZR2017MB061), the Taishan Scholar Project of Shandong Province of China (Nos. tsqn201812003 and ts20190908), the Qilu Youth Scholar Funding of Shandong University. Project for Scientific Research Innovation Team of Young Scholar in Colleges and Universities of Shandong Province (2019KJC028).

## Author contributions

Original idea was conceived by D.S., experiments and data analyses were performed by X.-Q.L., Y.-Z.L, Z.W., S.-S.Z., Y.-C.L., Z.-Y.G., and D.S., ESI-MS data were collected by L.F., circular dichroism data were collected by X.-Q.L., Z.W. and Z.-Z.C., luminescence data were collected by S.-S.Z. and Q.-W.X., DFT calculations data were collected by Y.-Z.L, structure characterizations were performed by X.-Q.L., Z.W., and D.S., manuscript was drafted by X.-Q.L., D.S., and C.-H.T. All authors discussed the results and commented on the manuscript.

## Competing interests

The authors declare no competing interests.
