## [Peer Review File · Nature Communications]

REVIEWER COMMENTS

Reviewer #1 (Remarks to the Author):

This is a very interesting paper by Sun and coworkers who describe the synthesis and enantioseparation of a face-rotating induced chiral Ag₁₄ nanocluster in an all-achiral environment. The characterization and evaluation of the chiral Ag₁₄ nanocluster with space group of non-enantiomorphous pairs (here P212121) needed dreaded workload, for example the terrible X-ray single crystal measurements (more than 40 crystals here!!). They picked crystals, collected the X-ray data, enriched the monochiral crystals to get convinced CD and CPL spectra. All the conclusions are solidly supported by the data. Especially, the excellent progresses in deciphering a novel chiral origin at molecular level and realizing the racemate to homochiral crystallization by the "competition" of the inter-cluster supramolecular interactions could justify publication in Nat. Commun. In my view, the manuscript is well organized, and the material selected for the main text and the supplementary information are scientifically presented. Overall this is an excellent and valuable contribution, thus I recommend the publication of after the authors address the following minor points:

- (1) The authors mentioned that large size single crystals can be obtained by adjusting the amount of triethylamine. Have the authors tried to use other kinds of bases in this system?
- (2) Considering the NaBH₄ as reducing agent, is there possibly some hydrides existed between core and shell? Can other reducing agent such as Ph₂SiH₂ work in this system?
- (3) There are some differences between experimental and calculated UV-VIS spectra, please give some possible explanations.
- (4) Ag₁₄ shows CD signals in the solid state, how about the CD spectrum in solution?
- (5) Some grammar errors should be taken care such as "indicated the excited states belongs to triplet excitation"

Reviewer #2 (Remarks to the Author):

The study by Sun and co-workers shows a detailed report on the synthesis and enantioseparation of face rotating induced chiral silver nanoclusters in an all-achiral environment, which deciphers the origin of chirality at molecular level.

The topic is highly important, and the presented results will surely deepen the understanding on the origin of chirality of clusters. The following are some comments that need minor revision:

- 1) The CD spectra of two chiral Ag₁₄ were given to verify the chirality of Ag₁₄, but the origin of their chirality was not further verified from the experimental characterization and DFT calculations. The CD signals combined the electron transitions (DFT calculation in your manuscript was given to analysis the relative transition of UV-vis spectra) might be helpful to verify the origin of chirality. (e.g., Angew. Chem. Int. Ed. 2018, 57,9059-9063)
- 2) I suggest the authors to cite references such as J. Am. Chem. Soc., 1990, 112, 5534; Chem. Soc. Rev., 2009, 38, 564 were suggested to be cited to support the pi-pi interactions between the exterior pntp- ligands (with distances ranging from 3.55 to 3.77 angstrom).

Reviewer #3 (Remarks to the Author):

Chiral monolayer-protected metal clusters are getting much attention recently due to promising applications in areas like chiral catalysis. There are numerous examples of chiral clusters reported and the current contribution adds to this. The authors report a Ag₁₄ cluster that is chiral in the solid state. Interestingly, the cluster shows spontaneous resolution of enantiomers upon crystallization. This has been reported before for metal clusters. Whether the cluster crystallizes as racemate or as separated enantiomers depends on the solvent in the current case. This is an interesting point.

However, in my opinion this is not sufficient for a publication in Nature Communications, also for the reasons detailed below.

-It seems that the cluster can be separated in its enantiomers only in the solid state - the authors should clarify this. What happens in solution? Is there still a CD signal? If the cluster is chiral and enantiopure only in the solid state it will be difficult to derive applications.

-The authors write: "Although some technologies such as high performance liquid chromatography (HPLC), capillary electrophoresis (CE) and chiral ion pairs have been developed to separate enantiomers, high price, complicated operation, low efficiency...." As I understand in the current case the crystals are picked one by one from the sample and then each crystal is analysed by single crystal X-ray diffraction. This seems comparably complicated compared to chromatographic separation!

-Why are the absorption spectra in Figure 6 and Figure 8 so different?

-CPL: It looks like the signal to noise is about 1 in Figure 7. (I assume the light grey traces are the original traces.) How did the authors arrive at the almost noise-free traces? Also the authors claim Glum is almost 1%. Why is the trace then so noisy?

-The authors claim that the cluster is neutral based on mass spectrometry...a neutral cluster cannot be detected in mass spectrometry. In any case it is difficult and sometimes dangerous to assign charge state of the original species from mass spectrometry.

Reviewer #1 (Remarks to the Author):

This is a very interesting paper by Sun and coworkers who describe the synthesis and enantioseparation of a face-rotating induced chiral Ag₁₄ nanocluster in an all-achiral environment. The characterization and evaluation of the chiral Ag₁₄ nanocluster with space group of non-enantiomorphous pairs (here $P2_12_12_1$) needed dreaded workload, for example the terrible X-ray single crystal measurements (more than 40 crystals here!!). They picked crystals, collected the X-ray data, enriched the monochiral crystals to get convinced CD and CPL spectra. All the conclusions are solidly supported by the data. Especially, the excellent progresses in deciphering a novel chiral origin at molecular level and realizing the racemate to homochiral crystallization by the “competition” of the inter-cluster supramolecular interactions could justify publication in Nat. Commun. In my view, the manuscript is well organized, and the material selected for the main text and the supplementary information are scientifically presented. Overall this is an excellent and valuable contribution, thus I recommend the publication of after the authors address the following minor points:

Response: We are pleased and excited by above positive comments on the novelty and significance of our study. We also believe that the revised manuscript improved quality thanks to your comments. In the following, we provide responses to the comments and suggestions point-by-point.

(1) The authors mentioned that large size single crystals can be obtained by adjusting the amount of triethylamine. Have the authors tried to use other kinds of bases in this system?

Response: Thanks for your constructive suggestion. This is really a good suggestion regarding to the systematical studies on the synthesis of Ag₁₄ nanocluster because the larger crystals will facilitate to study their CD and CPL properties. It is obligatory for us to try different kinds of bases in this assembly system in order to optimize the yield and size of crystals. For example, we have tried the other two bases: NaOH, and N,N,N',N'-Tetramethylethylenediamine (TMEDA) and adjusted their amounts in the synthesis of Ag₁₄ nanocluster. The results showed that Ag₁₄ nanocluster can also be obtained by replacing the

triethylamine with TMEDA. We can isolate the single crystals of the size of millimeters when 20 μL TMEDA were used. However, we cannot observe any crystals after switching to NaOH, it may be that reduction kinetics of NaBH_4 in the presence of NaOH is slower, which is not benefited for the formation of small silver nanoclusters, like Ag_{14} (*Acc. Chem. Res.* **51**, 1338–1348 (2018); *Angew. Chem. Int. Ed.* **53**, 4623-4627 (2014)). For clarity, we also added two sentences in main text as: Furthermore, we have tried the other two bases such as NaOH and N,N,N',N' -Tetramethylethylenediamine (TMEDA) in the synthesis of Ag_{14} nanocluster. The TMEDA can also work as triethylamine, whereas the NaOH isn't the same case, which may be due to the slower reduction kinetics of NaBH_4 in NaOH, impeding the formation of Ag_{14} nanocluster.

(2) Considering the NaBH_4 as reducing agent, is there possibly some hydrides existed between core and shell? Can other reducing agent such as Ph_2SiH_2 work in this system?

Response: Thanks for your constructive suggestion. It can be seen from the mass spectrometry that the most dominant peak (B) is intact molecular ion species but with only one sodium attachment thus assigned to $[\text{NaAg}_{14}(\text{pntp})_{10}(\text{dpnh})_4\text{Cl}_2]^+$. It shows that there is no hydrides existed in the interstice between core and shell of Ag_{14} nanocluster.

Fig. R1 Positive-ion mode ESI-MS of racemic conglomerates SD/L- Ag_{14} and SD/R- Ag_{14} dissolved in $\text{MeOH}/\text{CH}_2\text{Cl}_2$. Insets: Zoom-in ESI-MS of experimental (blue line) and simulated (red line) for each labeled species.

We also used Ph_2SiH_2 instead of NaBH_4 , and Ag_{14} nanocluster was not found under the same experiment conditions. This is most likely because NaBH_4 is a strong reducing agent, while Ph_2SiH_2 is relatively weaker, which has not strong enough reducibility to promote the formation of thermodynamically stable Ag_{14} nanocluster.

(3) There are some differences between experimental and calculated UV-vis spectra, please give some possible explanations.

Response: Thanks for your kind reminder. The absorption peak shift between experimental and calculated UV-Vis spectra is most probably attributed to the less accurate functional and/or basis sets adopted in the DFT calculation, which is often encountered in the UV-Vis simulation especially for large transition metal clusters. A scaling factor was thus not applied to diminish the shift in this case, as the electronic transition results are reasonable based on its specific electronic structure.

(4) Ag_{14} shows CD signals in the solid state, how about the CD spectrum in solution?

Response: Thanks for your constructive suggestion. In the crystalline phase, the vibration and rotation of the ligand can be effectively restricted and the non-covalent interactions based on $\pi \cdots \pi$ stacking interactions between pntp^- ligands lead to the asymmetric arrangement of the ligands, which induces the face-rotating of the Ag_8 cubic shell in Ag_{14} nanocluster to produce chirality. However, the vibration and rotation of the ligand become much more popular in the solution, the ligand configuration is dynamic and cannot maintain its asymmetrical arrangement. In other words, due to the fact that the origin of the chirality is dependent on $\pi \cdots \pi$ stacking interactions between pntp^- ligands, which can be labile in solution, thus allowing for the interconversion of enantiomers, the result leads to a racemic mixture (*Chem. Soc. Rev.*, 46, 2555-2576 (2017)), which can be verified by the silent circular dichroism (CD) signal in the solution (Fig. R2). This phenomenon was also observed in previous literatures (*Chem. Soc. Rev.*, 46, 2555-2576 (2017); *J. Am. Chem. Soc.* 132, 14 (2010); *Chem. Soc. Rev.*, 38, 830-845 (2009)).

Fig. R2 The CD and its corresponding absorption spectra of the one SD/L-Ag14 crystal dissolved in MeCN/CH₂Cl₂.

(5) Some grammar errors should be taken care such as “indicated the excited states belongs to triplet excitation”.

Response: Thanks for your careful checking. We have revised it in the main text according to your suggestions and marked red. Some other simialr errors were also revised by carefully checking the overall manuscript and SI.

Reviewer #2 (Remarks to the Author):

The study by Sun and co-workers shows a detailed report on the synthesis and enantioseparation of face rotating induced chiral silver nanoclusters in an all-achiral environment, which deciphers the origin of chirality at molecular level. The topic is highly important, and the presented results will surely deepen the understanding on the origin of chirality of clusters. The following are some comments that need minor revision:

Response: Thanks for your positive comments on our work. We agree with the reviewer that presenting the DFT calculations will provide strong evidence supporting the origin of chirality. The DFT calculations about CD were performed and some discussions were also added to enhance the quality of the manuscript. We believe that the revised manuscript improved quality thanks to your constructive suggestions.

1) The CD spectra of two chiral Ag₁₄ were given to verify the chirality of Ag₁₄, but the origin of their chirality was not further verified from the experimental characterization and DFT calculations. The CD signals combined the electron transitions (DFT calculation in your manuscript was given to analysis the relative transition of UV-vis spectra) might be helpful to verify the origin of chirality. (e.g., Angew. Chem. Int. Ed. 2018, 57, 9059-9063)

Response: Thanks for your constructive suggestions. Thanks for the important reminder about the corresponding DFT calculations on the CD spectra in main text. To further understand the origin of chirality, taking SD/R-Ag14 as an example, its metal core after DFT optimization in gas phase remains almost intact and thus its calculated CD spectrum can be used to help interpret the origin of its chirality. Although the CD curves above 500 nm showed less perfect mirror symmetry, two main shorter-wavelength Cotton effects at 359 nm and 417 nm strongly suggest their enantiomeric relationship. The calculated CD spectrum of SD/R-Ag14 matches fairly well with its experimental spectrum in the region below 500 nm (Fig. R3). The broad calculated absorption peak centered at 358 nm was split into three peaks by Cotton effect, including two medium positive peaks at 314 nm and 417 nm, and one strong

negative peak at 359 nm. All the CD responses are mainly associated with the LLCT transitions, despite the small proportion of MMCT and MLCT transitions, as illustrated in its UV-Vis absorption analysis (Supplementary Fig. 14 and table 6). This suggests that the chiral response in the Ag₁₄ nanocluster mainly originates from the asymmetrical arrangement of the surface ligands, which is induced by non-covalent interactions (mainly the $\pi\cdots\pi$ stacking interactions) between pntp⁻ ligands. The related discussions were also added into main text.

Fig. R3 Simulated (solid line) and experimental (dashed lines) CD spectra of SD/R-Ag14 nanocluster.

2) I suggest the authors to cite references such as J. Am. Chem. Soc., 1990, 112, 5534; Chem. Soc. Rev., 2009, 38, 564 were suggested to be cited to support the pi-pi interactions between the exterior pntp⁻ ligands (with distances ranging from 3.55 to 3.77 angstrom).

Response: Thanks for your constructive suggestion. We cited these two very important and closely related papers in the main text as refs 40-41.

Reviewer #3 (Remarks to the Author):

Chiral monolayer-protected metal clusters are getting much attention recently due to promising applications in areas like chiral catalysis. There are numerous examples of chiral clusters reported and the current contribution adds to this. The authors report a Ag_{14} cluster that is chiral in the solid state. Interestingly, the cluster shows spontaneous resolution of enantiomers upon crystallization. This has been reported before for metal clusters. Whether the cluster crystallizes as racemate or as separated enantiomers depends on the solvent in the current case. This is an interesting point. However, in my opinion this is not sufficient for a publication in Nature Communications, also for the reasons detailed below.

Response: We are pleased and excited by the reviewer's positive acknowledgement on the novelty and significance of our study. We fully understand the potential concerns from this reviewer, however, there are only few cases reported in this field. In previous cases, authors mainly focused on the report of a special chiral phenomenon or result instead of some underlying reasons for these phenomenon. Anyway, we deciphered the origin of chirality formed in polyhedral silver nanocluster and quickly achieved controllable conversion from racemates to pure enantiomer by identifying the special effects of intra- and inter-cluster supramolecular interactions. We would also like to thank the reviewer for constructive comments below, which have been taken into careful consideration in this revision. We believe that the revised manuscript improved quality thanks to your constructive suggestions. In the following, we provide concrete responses to the comments and suggestions point-by-point.

-It seems that the cluster can be separated in its enantiomers only in the solid state - the authors should clarify this. What happens in solution? Is there still a CD signal? If the cluster is chiral and enantiopure only in the solid state it will be difficult to derive applications.

Response: Thanks for your constructive suggestion. This important question was also raised by reviewer 1. Yes, the racemic Ag_{14} nanocluster can be separated in its enantiomers only in the solid state, which is decided by the effective chiral

configuration-locking effect of non-covalent interactions (mainly the $\pi\cdots\pi$ stacking interactions between pntp⁻ ligands) in solid state. In the crystalline phase, the vibration and rotation of the ligand can be effectively restricted, and the non-covalent interactions based on $\pi\cdots\pi$ stacking interactions between pntp⁻ ligands lead to the asymmetric arrangement of the ligands, which induces the face-rotating of Ag₈ cubic shell in the Ag₁₄ nanocluster to produce chirality. In the revision stage, we also performed DFT calculations on CD results which also suggest that the chiral response in the Ag₁₄ nanocluster mainly originates from the asymmetrical arrangement of the surface ligands. At the same time, these acetonitrile molecules in the crystal lattice, which can form C-H \cdots O/N hydrogen bonds with nitro group or dpnh on Ag₁₄ nanocluster, promoting the homochiral crystallization in solid state, finally the racemic Ag₁₄ nanocluster can be separated in its enantiomers.

However, the vibration and rotation of the ligand are not restricted in the solution, the ligand configuration is dynamic and cannot maintain its asymmetrical arrangement as seen in solid state. In other words, due to the fact that the origin of the chirality is dependent on $\pi\cdots\pi$ stacking interactions between pntp⁻ ligands, which can be labile in solution, thus allowing for the interconversion of enantiomers, the result leads to a racemic mixture (*Chem. Soc. Rev.*, 46, 2555-2576 (2017)), which can be verified by the silent circular dichroism (CD) signal in the solution (Fig. R2). This phenomenon was also observed in previous literatures (*Chem. Soc. Rev.*, 46, 2555-2576 (2017); *J. Am. Chem. Soc.* 132, 14 (2010); *Chem. Soc. Rev.*, 38, 830-845 (2009)).

Fig. R2 The CD and its corresponding absorption spectra of the one SD/L-Ag14 crystal dissolved in MeCN/CH₂Cl₂.

At present, circularly polarized luminescence (CPL) active materials have potential applications in the fields of 3D optical displays, information storage, molecular switches, asymmetric catalysis, biological probes, and spintronic devices, among which solid-state chiral silver nanoclusters have potential application in circularly polarized luminescence materials, such as organic light-emitting diodes (OLEDs). Generally, the luminous asymmetry factor (g_{lum}) in the solid state is larger than that in the solution (*Chem. Sci.* **10**, 843-847 (2019); *Sci. Adv.* **3**, e1700956 (2017); *Chem. Sci.* **3**, 2737–2747 (2012)), chiral substances with CPL properties are often derived applications in solid form (*Sci. Adv.* **6**, eaay0107 (2020); *J. Am. Chem. Soc.* **138**, 9743-9746 (2016); *Adv. Mater.* **27**, 1791-1795 (2015)), so solid CPL-active nanoclusters in application are highly expected. In our group and others, we are creating such kind of materials and enriching the family of such material library, which may provide candidates for application in the near future. -The authors write: “Although some technologies such as high performance liquid chromatography (HPLC), capillary electrophoresis (CE) and chiral ion pairs have been developed to separate enantiomers, high price, complicated operation, low efficiency....” As I understand in the current case the crystals are picked one by one

from the sample and then each crystal is analysed by single crystal X-ray diffraction. This seems comparably complicated compared to chromatographic separation!

Response: Thanks for your constructive suggestion. Yes, we fully agree with insightful suggestion about the description in the introduction that has been revised now. The high performance liquid chromatography (HPLC), capillary electrophoresis (CE) and chiral ion pairs have been developed to separate enantiomers, which are based on keeping a stable chiral configuration in solution, and additional chiral auxiliaries are needed to interact with the counterpart enantiomer to achieve enantiomer resolution. In another case, the chiral nanoclusters formed by the distortion or rotation induced by non-covalent interaction in an achiral environment cannot keep their stable chiral configuration in solution thus hardly to be separated by the above methods involving the solution treatment, so the spontaneous resolution may be a better choice to achieve enantiomer resolution. Based on the suggestion from editor Dr. Joan Serrano-Plana, we also toned down our claims related to the potential advantages of our methodology over other techniques for the separation of chiral compounds. Thus, we revised the related contents in main text as: Currently, some enantioseparation technologies such as high performance liquid chromatography (HPLC), capillary electrophoresis (CE) and chiral ion pairs have been developed to separate enantiomers, but there are still limitations to enantioseparation of most racemates. In contrast, we adopt a solvent-induced tactic to quickly achieve homochiral crystallization, then obtain enantiopure nanoclusters in the crystalline phase finally.³³⁻³⁸ Thus, spontaneous resolution may be an alternative choice to achieve enantiomer separation, especially for chiral molecules that cannot maintain a stable chiral configuration in solution.

-Why are the absorption spectra in Figure 6 and Figure 8 so different?

Response: Thanks for your constructive suggestion. Figure 8 are the UV-Vis absorption spectrum of Ag₁₄ nanoclusters in solution, while Figure 6 are the UV-Vis diffuse reflection spectrum of Ag₁₄ nanoclusters in solid state. Moreover, the testing methods and principles of solid and solution samples are different. The former collects the absorbance of the Ag₁₄ nanoclusters in solution, while the latter

collects the diffuse reflected signals of the Ag₁₄ nanoclusters solid sample. On the other hand, the UV-Vis spectrum for SD/L-Ag₁₄ or SD/R-Ag₁₄ in CH₂Cl₂ solution reflects the absorption behavior of single Ag₁₄ nanocluster, and the electronic transition is more refined, the electronic transitions contributed to absorption peak are mainly of LLCT (ligand-to-ligand charge transfer) character, that is the transition from p orbitals of S and C atoms in pntp⁻ ligands to the π* orbitals of C=C and N=O bonds in the pntp⁻ ligands. Apart from these, some MLCT (metal to ligand charge transfer) and MMCT (metal to metal charge transfer) transitions also contributed, so the peak shape is more sharper. By comparison, SD/L-Ag₁₄ or SD/R-Ag₁₄ in solid-state represents the absorption behavior of bulk sample (or Ag₁₄ nanoclusters packed together), where the electronic transition are more complicated because of the contribution of inter-molecular interactions, and simultaneous transitions between very close energy levels, so the peak shape is mostly broad, then they show some difference in UV-vis profiles.

-CPL: It looks like the signal to noise is about 1 in Figure 7. (I assume the light grey traces are the original traces.) How did the authors arrive at the almost noise-free traces? Also the authors claim Glum is almost 1%. Why is the trace then so noisy?

Response: Thanks for your constructive suggestion. The low signal to noise of CPL should arise from either the intrinsically weak fluorescence intensity of SD/L-Ag₁₄ or SD/R-Ag₁₄ or the low concentration of sample in compacted tablet during the CPL measurement. In order to deduct the noise, we smoothed the CPL original curves to make it closer to the true intensity of the CPL signal.

The luminous asymmetry factor (g_{lum}) was calculated by the formula $g_{lum}=2\times[CPL/(32\ 980/\ln 10)]/DC$ and its value also becomes not accurate because of the high noise signals. In addition, chiral lanthanide complexes have been shown to exhibit high values of g_{lum} because of the unique magnetic dipole transition nature of 4f electrons. In contrast, silver nanoclusters almost always exhibit much smaller levels of circular polarization ($g_{lum}< 0.01$) (*J. Am. Chem. Soc.* 140, 3683-3689 (2018); *Chem. Soc. Rev.*, 41, 7673-7686 (2012); *Angew. Chem. Int. Ed.* 51, 704-708 (2012); *Chem. Soc. Rev.*, 34, 1048-1077 (2005)). Therefore, the CPL signal will

inevitably be affected by noise, making signal detection difficult, especially when the concentration of sample is low.

Anyway, in order to address such concern from the reviewer and to obtain the reliable CPL data, we re-collected the enantiomer crystals and re-tested CPL and DC (Fig. R4). During the test, we also controlled the orientation of tablet samples with different rotation angles (0°, 90°, 180°, 270°) and tested each sample eight times at the same rotation angles to suppress the noise to get better signal to noise. The results are almost identical but with higher signal to noise and independent to rotation angles. Based on above measurements, the existence of linear polarized luminescence was also excluded, which ensures the possibility and reliability for CPL emergence (*Angew. Chem. Int. Ed.* 60, 3138-3147 (2021)). The revised CPL and DC spectra have been shown as follows and corrected in the main text.

Fig. R4 The CPL (a) and DC (b) spectra of SD/L-Ag14 or SD/R-Ag14 in the solid state under the excitation of 365 nm.

-The authors claim that the cluster is neutral based on mass spectrometry...a neutral cluster cannot be detected in mass spectrometry. In any case it is difficult and sometimes dangerous to assign charge state of the original species from mass spectrometry.

Response: Thanks for your constructive suggestion. The Ag₁₄ nanocluster is indeed neutral, but the Ag₁₄ nanocluster can carry charge by adding a sodium ions (Na⁺) from the glassware or transfer lines within the instrument during the

measurement of ESI-MS. As the solvent gradual evaporation in the ESI process, the Na^+ in the droplet has stronger interactions with the Ag_{14} nanocluster by electrostatic interactions or π -cation interactions, the Na^+ will tend to attach themselves to the Ag_{14} nanocluster, forming quasimolecular ions with positive charges then detected by mass spectrometry. In addition, there are several relevant documents that can fully prove such case, such as *Angew. Chem. Int. Ed.* 59, 2309-2312(2020); *Angew. Chem. Int. Ed.* 58, 11967-11977(2019); *J. Am. Chem. Soc.* 138, 10425-10428 (2016).

REVIEWER COMMENTS

Reviewer #1 (Remarks to the Author):

The concerns have been addressed and the reviewer now recommends publication.

Reviewer #2 (Remarks to the Author):

The authors have adequately settled my concerns, and I would be happy to approve its publication now.

Reviewer #3 (Remarks to the Author):

The authors have addressed my points and I am quite satisfied with the answers and the changes made to the manuscript. I proposed publication of the manuscript.

RESPONSE TO REVIEWERS' COMMENTS:

Reviewer #1 (Remarks to the Author):

The concerns have been addressed and the reviewer now recommends publication.

Response: We appreciate the reviewer's recommendation very much.

Reviewer #2 (Remarks to the Author):

The authors have adequately settled my concerns, and I would be happy to approve its publication now.

Response: We sincerely thank the reviewer for the positive comment.

Reviewer #3 (Remarks to the Author):

The authors have addressed my points and I am quite satisfied with the answers and the changes made to the manuscript. I proposed publication of the manuscript.

Response: Thank you very much. We appreciate the reviewer positive recommendation for publication of our work in Nature Communications.